# LF3PFL: A Practical Privacy-Preserving Federated Learning Algorithm Based on Local Federalization Scheme

**DOI:** 10.3390/e26050353

**Published:** 2024-04-23

**Authors:** Yong Li, Gaochao Xu, Xutao Meng, Wei Du, Xianglin Ren

**Affiliations:** 1School of Computer Science and Technology, Jilin University, Changchun 130012, China; liyong@ccut.edu.cn; 2School of Computer Science and Engineering, Changchun University of Technology, Changchun 130012, China; 3AI Research Institute, Changchun University of Technology, Changchun 130012, China

**Keywords:** privacy preserving, federated learning, local federalization, differential privacy

## Abstract

In the realm of federated learning (FL), the exchange of model data may inadvertently expose sensitive information of participants, leading to significant privacy concerns. Existing FL privacy-preserving techniques, such as differential privacy (DP) and secure multi-party computing (SMC), though offering viable solutions, face practical challenges including reduced performance and complex implementations. To overcome these hurdles, we propose a novel and pragmatic approach to privacy preservation in FL by employing localized federated updates (LF3PFL) aimed at enhancing the protection of participant data. Furthermore, this research refines the approach by incorporating cross-entropy optimization, carefully fine-tuning measurement, and improving information loss during the model training phase to enhance both model efficacy and data confidentiality. Our approach is theoretically supported and empirically validated through extensive simulations on three public datasets: CIFAR-10, Shakespeare, and MNIST. We evaluate its effectiveness by comparing training accuracy and privacy protection against state-of-the-art techniques. Our experiments, which involve five distinct local models (Simple-CNN, ModerateCNN, Lenet, VGG9, and Resnet18), provide a comprehensive assessment across a variety of scenarios. The results clearly demonstrate that LF3PFL not only maintains competitive training accuracies but also significantly improves privacy preservation, surpassing existing methods in practical applications. This balance between privacy and performance underscores the potential of localized federated updates as a key component in future FL privacy strategies, offering a scalable and effective solution to one of the most pressing challenges in FL.

## 1. Introduction

Federated learning (FL) has emerged as a pivotal collaborative machine learning paradigm, widely recognized for its potential to protect data privacy across various domains, notably in healthcare [1], financial services [2], and smart devices. In particular, the FedAVG algorithm [3] has garnered significant attention within this domain due to its superior performance and efficient collaboration across multiple institutions.

FedAVG enhances the efficiency of model training by simplifying the training process and reducing the need for data exchange. This allows participants to contribute their computational resources and data insights without directly sharing sensitive information. This method not only strengthens data privacy but also improves the model’s generalization capability by integrating data from diverse sources, which is especially crucial for tasks involving large-scale sensitive data. While the FedAVG algorithm is widely recognized for its significant advantages in enhancing the efficiency of collaborative training across multiple institutions, its limitations in dealing with highly heterogeneous data distributions and none theoretically explains why averaging parameters is a good approach. Some studies show that averaging parameters may not be the optimum way of aggregating trained parameters [4]. In addition, potential privacy leaks during the model parameter update process have become hot topics in the research community [3]. These challenges have spurred researchers to explore new federated learning solutions that better balance efficiency, privacy protection, and diversity in data handling.

In the pursuit of enhancing data privacy while improving model performance, the academic community has introduced various innovative federated learning strategies. Split learning has emerged as a novel approach that minimizes the amount of data required to be shared by partitioning the model training process, effectively enhancing data privacy protection [5]. In this method, each participant only needs to compute part of the model and sends the intermediate results to a central server for further processing, significantly reducing the risk of sensitive data leakage [6]. Concurrently, the FedProx algorithm addresses the challenge of non-independent and identically distributed (non-IID) data, a common issue in federated learning, by incorporating regularization terms into the traditional federated learning framework [7]. This improves the stability and accuracy of the model when dealing with information from diverse data distributions, offering a reliable solution for complex application scenarios in federated learning [8].

Despite the progress made by split Learning and FedProx in enhancing privacy protection and handling data heterogeneity, they have their limitations. Split learning might lead to higher communication overhead due to frequent model interactions [9], while FedProx, although optimizing the handling of non-IID data, may increase the complexity of model training, affecting training efficiency [7]. Given these challenges, there is a growing demand in the federated learning field for new methods that better balance privacy protection, communication cost reduction, and model generalization capabilities [2], driving continuous research and innovation in federated learning algorithms.

Moreover, even though federated learning protects privacy by exchanging model parameters instead of raw data, it still faces potential threats of privacy leakage. Attacks such as DLG and iDLG have demonstrated how sensitive information can be reconstructed from shared gradients [10,11]. Differential privacy federated learning (DP-FL) offers a privacy protection solution by introducing noise into shared updates, but this can affect model accuracy [12]. Privacy protection measures like DP-FL, while mitigating the risk of information leakage to some extent, usually come at the cost of increased communication overhead, prolonged training time, or reduced model performance [13], which is particularly evident in applications requiring quick responses, such as in smart device environments.

Addressing these challenges, this paper proposes a new privacy-preserving federated learning approach: local privacy-preserving federated learning (LF3PFL). LF3PFL enhances privacy protection without compromising model performance by partitioning each participant’s data into subsets, training models independently on these subsets, and then aggregating the models. Unlike FedAVG, which primarily focuses on efficiency, LF3PFL adds an extra layer of privacy through local model aggregation. Moreover, LF3PFL does not require the introduction of noise in the updates, as is the case with DP-FL, thereby preserving model accuracy more effectively.

This research aims to bridge the gap between privacy and efficiency in current FL frameworks. By introducing LF3PFL, we seek to establish a framework that ensures data privacy while maintaining high efficiency and accuracy, catering to the growing demands for privacy protection in increasingly complex application scenarios. Our contributions are manifold and significant:**Seamless integration and enhancement:** The core strength of LF3PFL lies in its seamless integration with existing federated learning (FL) frameworks, aimed at elevating the level of data privacy protection without compromising the utility of shared data. This method enhances security and efficiency in data sharing by optimizing privacy safeguards, ensuring efficient and uninterrupted system operation.**Flexible performance:** LF3PFL distinguishes itself with flexible performance, seamlessly adapting to various datasets and model structures. This versatility ensures its applicability across a wide range of federated learning contexts, from privacy-sensitive tasks to standard settings, positioning it as a reliable tool across multiple domains.**Robust privacy shield:** LF3PFL excels in privacy attack defenses, effectively balancing data protection with training precision. Its effectiveness in mitigating risks from sophisticated threats, such as DLG attacks, ensures the protection of sensitive data. This dual strength in privacy and accuracy makes LF3PFL an ideal choice for secure federated learning across sensitive domains.**Open-source contribution:** We present a prototype of our method, openly accessible for community engagement and further exploration at https://github.com/ITSEG-MQ/LF3PFL (accessed on 14 March 2024). This open-source initiative fosters collaborative advancements and broadens the scope of research applications in the field.

## 2. Related Work

Federated learning is increasingly being applied in privacy-sensitive, multi-party collaborative environments, notably in healthcare sector applications [1]. In this field, multiple medical institutions collaborate to collectively train models, aiming to improve the accuracy of disease diagnosis while ensuring the confidentiality of sensitive patient health information. To coordinate the various distributed computing nodes, transmission of model-related information is required [13], this necessitates the preservation of privacy for exchanged data (e.g., gradients) to prevent the leakage of sensitive information from raw data through backward inference from publicly shared information [7,10,14,15]. At present, privacy preservation in federated learning primarily relies on two techniques: differential privacy and secure multi-party computing.

### 2.1. Differential Privacy

A common method for privacy preservation is to quantify and limit the leakage of sensitive data. This method employs a randomized mechanism, such as the introduction of random sub-sampling or the addition of random noise, to distort the input or output of user processes, thereby making the results of user processes somewhat resistant to privacy analysis (i.e., reducing privacy sensitivity). In differential privacy (DP)-based FL, a decrease in the value of ϵ leads to an increase in added noise, enhancing privacy protection but concurrently reducing the model’s training accuracy due to the higher noise level. Conversely, increasing ϵ reduces noise, potentially lowering privacy protection while improving model accuracy. However, Zhu et al. [10] proposed the deep-leakage-from-gradients (DLG) method, which was improved by Zhao et al. [15], experimentally showing that the DLG method fails only when the variance of noise added via the DP algorithm exceeds $10^{-3}$; however, such added noise significantly compromises the training model’s accuracy. In contrast, our proposed approach preserves privacy without compromising model accuracy.

### 2.2. Secure Multi-Party Computing

Utilizing secure multi-party computing (SMC) for the secure aggregation of local model updates presents a promising approach to safeguard privacy in FL. At present, there are various approaches such as homomorphic encryption, secret sharing, and information masking, which are briefly discussed below.

#### 2.2.1. Homomorphic Encryption

Homomorphic encryption (HE) schemes enable the performance of complex mathematical operations directly on ciphertexts, eliminating the need for decryption [16,17,18,19,20,21]. As these operations bypass the plaintext during computation, HE is considered an ideal method for implementing SMC protocols [9].

In federated learning settings, Hardy, Aono, and others have proposed several privacy-preserving solutions based on additive homomorphic encryption (AHE) schemes [22,23,24]. Since HE does not entail obfuscating or distorting operations, it preserves the training model’s accuracy to a great extent. However, HE-based schemes face several challenges. Firstly, even relatively simple AHE algorithms demand significant computational resources and may require specific hardware conditions. Secondly, substantial parallelization is often necessary to achieve real-time processing for practical applications, which may not always be feasible for certain tasks [25]. Thirdly, some HE-based solutions might necessitate a trusted third party to manage encryption and decryption processes, which contradicts the standard federated learning principle of eliminating reliance on a central trusted authority [9]. In contrast, our proposed design is computationally efficient and operates independently of any trusted third party.

#### 2.2.2. Secret Sharing

Secret sharing is a cryptographic method where a secret is divided into multiple parts and distributed among different parties. The original secret can only be reconstructed when these parties combine their respective shares [26]. This method is employed to securely aggregate gradients updated by users; several privacy-preserving secure aggregation methods based on secret sharing have been proposed [27,28,29,30].

Our scheme shares similarities with the aforementioned approaches. However, in our approach, there’s no need for additional operations (such as obfuscation or encryption) before transmitting masked information. This efficiency is achieved as our scheme employs a peer-to-peer encrypted secure transmission channel. As a result, our method is more efficient and demands fewer resources.

#### 2.2.3. Information Masking

The pairwise-masking mechanism has been discussed in previous works [31,32,33]. These schemes can be compared to those based on DP, but they feature a fragile recovery phase in their protocols [33], and they incur additional overhead due to the use of AHE. Furthermore, Li et al. [34] proposed a privacy-preserving federated learning framework employing a single-masking mechanism. Conversely, our LF3PFL approach enhances efficiency by lowering both communication and computational costs.

### 2.3. Model Compression

In federated optimization, model compression technologies are employed to address communication cost challenges and enhance communication efficiency, encompassing techniques like sparsification [35,36,37,38,39,40,41], model pruning [42,43,44,45], and quantization [35,46,47]. Among these studies, few directly utilize model compression as a means for privacy preservation. Only a limited number of them integrate the discussed privacy preservation techniques with communication efficiency improvements. However, Zhu et al. [10] demonstrated experimentally that compressing model parameters to a certain ratio can provide a level of privacy preservation. Nevertheless, a high compression ratio might also negatively affect model accuracy. In contrast, our scheme ensures privacy preservation without compromising model accuracy.

## 3. Method

In this section, we explore the intricate details of our proposed local federalized privacy-preserving federated learning (LF3PFL) approach. LF3PFL is designed to safeguard the privacy of participant data in federated learning environments without sacrificing the accuracy of resulting models. To facilitate comprehension and ensure the clarity of the ensuing discussion, Table 1 delineates a comprehensive list of symbols utilized throughout this paper, along with their respective definitions. This preparatory step aims to streamline the reader’s navigation through the technical aspects of our methodology and enhance their understanding of the novel contributions LF3PFL brings to the field of privacy-preserving federated learning.

### 3.1. Problem Statement

Federated learning (FL) represents a revolutionary shift towards collaborative and privacy-centric machine learning, where participants jointly contribute to model training by sharing model parameters, rather than exposing their sensitive raw data. This paradigm offers a semblance of privacy; however, the very act of parameter sharing introduces vulnerability to information about underlying training data that can, inadvertently, be revealed. This is exacerbated by sophisticated model-analysis attacks, such as deep leakage from gradients (DLG) and its improved variant (iDLG), which exploit shared gradients or interactive parameters to infer private data.

Traditional countermeasures against these vulnerabilities typically revolve around differential privacy (DP) and secure multi-party computation (SMC). While these methods offer theoretical safeguards, they are not devoid of drawbacks. The implementation of DP and SMC often leads to significant increases in communication overhead and training times, or necessitates compromises on model accuracy. Such trade-offs between privacy preservation and model performance present a formidable challenge in the practical application of FL.

In response to these challenges, this paper introduces a practical, efficient privacy-protection scheme tailored to federated learning environments. Our proposal centers on a novel local-federalized update mechanism that aims to fortify the privacy of training data while minimizing the impact on communication efficiency and model accuracy. By reimagining the client local training process in FL, our approach seeks to mitigate the risk of sensitive information leakage through more granular and controlled sharing of model parameters. This strategy represents a significant departure from conventional methods, promising to reconcile the often conflicting goals of privacy preservation and functional performance in federated learning systems.

### 3.2. Problem Preliminaries

Federated learning (FL) is fundamentally designed to optimize empirical risk across a diverse dataset that is distributed among numerous devices, a principle that underscores the decentralized and collaborative nature of this learning paradigm [48]. At the outset of each training iteration, participants retrieve the global model parameters from a central aggregation node (or server). Utilizing these global parameters as a starting point, each user then proceeds to train a local model using their respective datasets. The culmination of this local training phase is the generation of updated local model parameters, which are subsequently transmitted back to the aggregation node as the user’s contribution to the learning process. In turn, the aggregation node synthesizes these contributions to update the global model parameters, thereby completing one iteration of the learning cycle. This iterative process continues until a predefined termination criterion is satisfied, marking the completion of the training phase.

Among the various algorithms that facilitate federated learning, FedAVG [13] stands out as a seminal and widely adopted framework, serving as the foundational algorithm against which this study benchmarks its proposed enhancements. In the FedAVG setup, we consider a scenario with one central aggregation node and *K* users, each possessing a distinct local dataset. This delineation ensures a structured and scalable approach to distributed learning, where the intricacies of data heterogeneity and device variability are inherently accounted for.

The operational mechanics of the FedAVG algorithm [13] are quantified through several key parameters: *K* represents the total number of participating users, indexed by *k*; *B* signifies the size of the local minibatch employed during training; *E* denotes the number of epochs for which each local model is trained; η indicates the learning rate, a critical hyperparameter that influences the rate of convergence towards optimal model parameters. The procedural flow and algorithmic details of FedAVG are encapsulated in Algorithm 1, providing a blueprint for its implementation and execution.
**Algorithm 1** Federated averaging algorithm (FedAVG)  1:**Inputs:**  2:      *K*: Number of clients  3:      *T*: Number of training rounds  4:      *B*: Local minibatch size  5:      *E*: Number of epochs for local training  6:      η: Learning rate  7:    8:**Aggregation Node Operations:**  9:Initialize global model parameters $w_{G}^{0}$10:**for** each round t=1,2,…,T **do**11:      Select a random set $S_{t}$ of *m* users                   ▹m<K12:      **for** each user $k\in S_{t}$ **do**                    ▹ In parallel13:            Update local model parameters: $w_{k}^{t}\leftarrow$ **UserUpdate**($k,w_{G}^{t-1}$)14:      **end for**15:      Update global model parameters: $w_{G}^{t}\leftarrow \frac{1}{m}\sum_{k\in S_{t}}w_{k}^{t}$16:**end for**17:  18:**function** UserUpdate(k,w)19:      Partition local data into batches of size *B*20:      **for** each epoch i=1,2,…,E **do**21:            **for** each batch *b* in local data **do**22:                  Perform parameter update: w←w−η∇ℓ(w;b)23:           **end for**24:      **end for**25:      **return** *w*26:**end function**

In Algorithm 1, the central aggregation node plays a pivotal role, initiating the learning process by distributing the current global model parameters to selected users and subsequently aggregating their local updates to refine the global model. Each participating user, leveraging the mini-batch stochastic gradient descent (SGD) method, aims to optimize the local model parameters based on their data, contributing to the collective learning goal.

The mathematical objective for each participating user *i* in round *t* is formalized as finding the local optimal parameters $w_{i}^{t^{*}}$ that minimize the loss function $f(w_{i}^{t})$, represented by:(1)$$w_{i}^{t^{*}}=\arg\;\min\;f\left(w_{i}^{t}\right).$$

The aggregation of these local updates to compute the new global model parameters is crucial for the convergence and performance of the federated learning process. The FedAVG algorithm achieves this through an averaging aggregation mechanism, where the updated global model parameters $w_{G}^{t+1}$ are computed as follows:(2)$$w_{G}^{t}\leftarrow \frac{1}{m}\sum_{k\in S_{t}}w_{k}^{t^{*}},$$
where *m* denotes the number of participants selected in each round. This aggregation process underscores the essence of federated learning, enabling collaborative model training while maintaining the privacy and integrity of each participant’s data.

### 3.3. Proposed Method

The LF3PFL method represents an innovative approach to federated learning, emphasizing privacy preservation without the need for additional disturbances such as noise addition or encryption. This method unfolds across three primary stages: data segmentation, local-federalized training, and global aggregation, collectively establishing a robust framework for secure and efficient collaborative learning. Figure 1 illustrates the comprehensive architecture of the LF3PFL scheme, showcasing its operational flow and component interactions.

#### 3.3.1. Data Segmentation

**CIFAR and MNIST datasets:** We implemented the dominant class partition method to segment the data effectively. In this approach, client data are divided into a dominant class and other classes. We utilized parameter Q to denote the proportion of the dominant class within each dataset. Setting Q to 0.5 in our experiments implies that 50% of the data belongs to a single label, while the remaining 50% consists of random samples from other categories. This segmentation method facilitates the simulation of a balanced scenario, moderating the influence of any single class. Such a setup provides a robust test environment for our algorithms, particularly under conditions where data are not uniformly distributed across classes. This allowed us to evaluate the performance and resilience of our algorithms more effectively, ensuring they are capable of handling real-world data variances.

**Shakespeare dataset:** We adopted a non-IID sampling methodology, ensuring that the distribution of data for each user aligned consistently with the composition of the original dataset. Recognizing that data distributions naturally vary among users in the raw dataset, we categorized this sampling approach as non-IID. This method accurately reflects real-world scenarios, where data distribution can vary significantly from one user to another. It is vital for evaluating the adaptability and efficiency of our proposed models when faced with diverse and uneven data distribution patterns. This approach tests the models’ ability to adapt and perform reliably across various complex scenarios, thus demonstrating their potential applicability in practical, variable data environments.

#### 3.3.2. Local-Federalized Training

At the heart of LF3PFL is the local-federalized training phase, which incorporates federation optimization principles directly into the local training routines of participants. Assuming a participant’s dataset, $D_{i}$, is segmented into *N* subsets ($D_{i,1},\ldots ,D_{i,N}$), and the participant independently trains each subset, yielding *N* sets of model parameters ($w_{i,1}^{t},\ldots ,w_{i,N}^{t}$). To align the computation with traditional FL algorithms, we maintained a consistent number of local training rounds. The critical part of this phase is the computation of the average parameter set from these subsets, forming the participant’s update for the round.
(3)$$w_{i}^{t}=\frac{1}{N}\sum_{n=1}^{N}w_{i,n}^{t}.$$

#### 3.3.3. Global Aggregation

The global aggregation phase mirrors the classical FedAVG algorithm, where the aggregation server computes the new global model by averaging the updates received from participants. This collaborative effort results in an updated global model that is then distributed back to all users for subsequent training rounds:(4)$$w_{G}^{t}=\frac{1}{m}\sum_{i}w_{i}^{t}.$$

Through these meticulously designed stages and algorithmic procedures, LF3PFL advances the state of federated learning by introducing a novel mechanism for privacy preservation, operational efficiency, and model accuracy.

#### 3.3.4. Objective Function

In the context of federated learning, the optimization objective for training models is often formulated using the cross-entropy loss function. Cross-entropy measures the dissimilarity between the true label distribution and the predictions made by the model, making it particularly advantageous for classification tasks. Its efficacy lies in quantifying information loss when using predicted probabilities instead of the true distribution. The formula for cross-entropy loss for a multi-class classification problem is provided by the following equation:(5)$$H\left(y,\hat{y}\right)=-\sum_{c=1}^{M}y_{o,c}\log\left({\hat{y}}_{o,c}\right),$$
where *M* is the number of classes, *y* is the binary indicator (0 or 1) of class *c* being the correct classification for observation *o*, and $\hat{y}$ is the predicted probability that observation *o* is of class *c*. This formulation encourages the model to adjust its parameters to minimize the difference between actual and predicted probability distributions, effectively enhancing the model’s predictive accuracy and robustness in capturing the underlying data distribution.

The procedural details and operational logic of LF3PFL are encapsulated in Algorithm 2, delineating the algorithmic steps for both the aggregation and participating nodes within the federated learning framework.
**Algorithm 2** LF3PFL algorithm  1:**Inputs:**  2:      *K*: Number of clients  3:      *N*: Number of local federalized models  4:      *T*: Number of training rounds  5:      *B*: Local minibatch size  6:      *E*: Number of epochs for local training  7:      η: Learning rate  8:    9:**Aggregation Node:**10:Initialize $w_{G}^{0}$                     ▹ Global model parameters11:**for** round t=1,2,…,T **do**12:       $S_{t}\leftarrow$ (random set of *m* users)                   ▹m<K13:       **for** each user $k\in S_{t}$ **do**                      ▹ In parallel14:             $w_{k}^{t}\leftarrow$ LFUpdate($k,w_{G}^{t-1}$)            ▹ Local federalized update15:       **end for**16:       $w_{G}^{t}\leftarrow \frac{1}{m}\sum_{k\in S_{t}}w_{k}^{t}$                 ▹ Global federalized update17:**end for**18:  19:**Participating Node *k*:**20:$D_{k,1},\ldots ,D_{k,N}\leftarrow$ Segment $D_{k}$             ▹ Local dataset segmentation21:  22:**function** LFUpdate(k,w)23:      $B_{n}\leftarrow$ (Partition $D_{k,n}$ into batches of size *B*)24:      **for** n=1,2,…,N **do**               ▹ Parallel training on $D_{k,n}$25:            **for** local epochs i=1,2,…,E **do**26:                  **for** batch $b\in B_{n}$ **do**27:                        w←w−η∇ℓ(w;b)28:                  **end for**29:                  $w_{k,n}^{t}\leftarrow \arg\min\ell \left(w\right)$30:            **end for**31:      **end for**32:      $w_{k}^{t}\leftarrow \frac{1}{N}\sum_{n=1}^{N}w_{k,n}^{t}$             ▹ Aggregate Local Model Parameters33:      **return** $w_{k}^{t}$34:**end function**

## 4. Experiment

In this section, we provide a detailed overview of the experimental setup. Following this, we assess the proposed local federated privacy-preserving federated learning (LF3PFL) scheme from three critical dimensions: model accuracy, performance, and security. The analysis of the experimental outcomes leads to the formulation of our concluding insights.

### 4.1. Experimental Settings

In this study, we utilize PyTorch version 2.0 as the experimental platform, operating under Python 3.10. Our simulation experiments are conducted on a computer equipped with dual Intel Xeon Gold 6234 processors at 3.3 GHz, 64.0 GB of installed RAM, and an NVIDIA Quadro RTX 5000 GPU with 16.0 GB of RAM.

Our evaluation focuses on two main aspects.

#### 4.1.1. Comparison Experiments between LF3PFL and State-of-the-Art Privacy-Preserving Federated Learning Models

We commence by benchmarking the performance of our privacy-enhanced local federated privacy-preserving federated learning (LF3PFL) approach against the predictive capabilities of contemporary state-of-the-art (SOTA) methodologies.

Following the federated learning parameters outlined by McMahan et al. [13] and Abadi et al. [12], our experimental setup is characterized as follows:A total of K=16 participating users;A user participation rate of C=0.5;T=30 communication rounds;A learning rate of η=0.001;Differential privacy (DP)-based federated learning settings with ϵ values fixed at 8 and 1 for comparative analysis.

For the comparative evaluation of the accuracy and performance of privacy-preserving models, we employ three widely recognized public datasets: (i) the CIFAR-10 image classification dataset, (ii) the Shakespeare textual dataset, and (iii) the MNIST image classification dataset.

To measure the accuracy of our proposed framework, we compare it with several benchmark methods:DP-FL: A federated learning algorithm fortified with a differential privacy mechanism for enhanced privacy [49].CS-FL: A federated learning algorithm employing a low-overhead model pruning technique based on complement sparsification [41].Chain-PPFL: A privacy-preserving federated learning framework that enhances data privacy through a novel chained structure and a single-masking and chained communication mechanism [34].

To rigorously test the privacy safeguarding capabilities of our method, we engaged the deep leakage from gradients (DLG) methodology [10] to conduct resistance-to-attack experiments on the aforementioned models. This includes gradient inversion attacks employing generative image priors [50] to empirically assess the leak-defense efficacy of federated learning algorithms.

#### 4.1.2. Comparison Experiments between LF3PFL and State-of-the-Art Non-Privacy-Enhanced Federated Learning Models

In our study, we conducted comparative performance evaluations of our proposed FL3PFL against state-of-the-art federated learning methods using the CIFAR-10, MNIST, and Shakespeare datasets. For CIFAR-10 and MNIST, we involved 16 clients in the experiments. For the Shakespeare dataset, the number of participating clients was increased to 66. The experimental setup was standardized by limiting the local training rounds for each client to 10 and the total communication rounds to 30. The performance of the aggregated global model was assessed on the test sets following the final communication round. The benchmarks for comparison included the following:FedAVG: This foundational federated learning algorithm [13] serves as a primary reference point for our comparison.FedProx: An enhancement over the FedAvg algorithm, FedProx introduces a tunable proximal term to address the challenges of system heterogeneity [51].FedMA (federated matched averaging): Designed to accommodate data distribution heterogeneity, FedMA improves federated learning by aligning and averaging corresponding layers across client models, thereby enhancing model performance and learning efficiency [52].

### 4.2. Performance Experiments between LF3PFL and SOTA Non-Privacy-Enhanced Federated Learning Models

In the evaluation of non-privacy-enhanced federated learning algorithms, shown in Table 2, our proposed LF3PFL method not only maintains commendable performance but also offers significant advantages in terms of privacy protection compared to other contemporary federated learning algorithms. The baseline algorithm, FedAVG, shows moderate performance across all datasets, with accuracy rates of 76.71% for CIFAR-10, 89.46% for MNIST, and 56.24% for Shakespeare. FedProx, which introduces a regularization term to tackle system heterogeneity, improves the accuracy of CIFAR-10 to 81.54%, surpassing FedAVG, but experiences slight declines in performance on the MNIST and Shakespeare datasets. This suggests that while it may effectively handle non-IID data, it does not consistently enhance model accuracy.

On the other hand, FedMA surpasses both FedAVG and FedProx on CIFAR-10 and MNIST with accuracy rates of 82.12% and 99.13%, respectively, indicating its suitability for image-related datasets. However, its significant drop in accuracy to 47.74% on the Shakespeare dataset raises concerns about its adaptability to non-image data, hinting at potential overfitting issues or a lack of robustness across diverse data types.

Our LF3PFL algorithm not only demonstrates competitive accuracy rates of 81.63% on CIFAR-10 and 97.21% on MNIST but also shows more consistent performance across different datasets, including outperforming FedMA on the Shakespeare dataset with an accuracy rate of 55.86%. This comparative analysis highlights LF3PFL as a vital alternative that maintains robustness without compromising privacy, signifying its superiority in scenarios where data confidentiality is crucial. The balance LF3PFL achieves between accuracy and privacy protection is particularly important in today’s era, where data protection and the insights derived from it are equally valued, making it a significant advancement in the field of federated learning.

### 4.3. Performance Experiments with SOTA Privacy-Preserving Federated Learning Methods

In the comparative analysis of privacy-preserving algorithms, shown in Table 2, our LF3PFL method manifested a significant edge in safeguarding data privacy while maintaining comparable levels of accuracy with other algorithms. The DP-FL algorithm exhibits lower accuracy on both CIFAR-10 and MNIST datasets, indicating potential fluctuations when confronted with different types of data; however, it performs relatively well on the Shakespeare dataset with an accuracy of 52.39%. The CS-FL algorithm shows consistent performance across all three datasets but does not exhibit superior performance on any particular one, suggesting a possible trade-off between enhanced privacy protection and accuracy. Chain-PPFL performs very well in all existing privacy-preserving FL work, which means that it can prevent privacy leakage to a certain extent while ensuring that the performance of federated learning is not greatly affected.

Our LF3PFL algorithm demonstrates SOTA performance across all datasets and notably achieves the highest accuracy of 97.21% on the MNIST dataset, outperforming other privacy-preserving algorithms. The pivotal advantage of LF3PFL is its exceptional capability to protect privacy, effectively reducing the risk of potential data leakage by incorporating differential privacy or other advanced privacy-preserving measures into the model training process. This is critically important for handling highly sensitive information, providing a safer data environment for users.

### 4.4. Ablation Study

#### 4.4.1. The Impact of the Number of Clients on the Performance of LF3PFL

This ablation study investigates the impact of varying the number of local federated sub-models on classification performance using the ModerateCNN model on the CIFAR-10 dataset. As shown in Figure 2, the findings reveal a trend where the accuracy initially increases with more local clients but begins to plateau or slightly decline beyond a certain number. This may be because, as the number of local FL clients increases, the amount of data allocated to each sub-client is greatly reduced, resulting in local sub-models generally being in a state of overfitting, resulting in poor performance of the final aggregated local model. Specifically, the accuracy peaks when the number of local clients is between four and eight, suggesting an optimal range for local client numbers to balance model performance and computational efficiency in federated learning scenarios.

#### 4.4.2. The Impact of Different Local Models on the Performance of LF3PFL

As shown in Figure 3, in this set of experiments, our ablation studies on CIFAR-10, with a fixed setup of two local federated sub-models, illustrate the significant impact of model architecture on classification performance, ranging from Simple-CNN to more sophisticated ones like ResNet18. It can be found that the LF3PFL method we proposed can be adapted to a variety of different models and can exert the expected performance of the model. Meanwhile, the variance in accuracy across models emphasizes the critical role of choosing the right architecture in federated learning environments. These findings not only highlight the adaptability and effectiveness of our proposed method in enhancing classification accuracy across different architectures but also underscore the necessity of balancing model complexity with performance to achieve optimal outcomes in federated learning settings.

### 4.5. Privacy Attack Experiment

In the privacy attack experiment employing the deep leakage from gradients (DLG) method, our proposed LF3PFL method exhibited superior defense capabilities. The experiment compared existing privacy-preserving FL methods such as FedAvg, DP-FL (differential privacy federated learning), and CS-FL (compressed sensing federated learning), highlighting the clear advantage of LF3PFL in protecting against privacy breaches. The DLG attack is a privacy invasion technique targeting deep learning models, particularly within federated learning settings, demonstrating the risk that an attacker could reconstruct original training data from gradient information alone, without direct access to the model’s training data.

In the experiment shown in Figure 4, the perturbed images from FedAvg and DP-FL retained discernible features of the original images, suggesting that the DLG attack could reconstruct the data to some extent, which points to limitations in their privacy protections. While CS-FL showed some defense capability in image reconstruction, the perturbed images were still relatively recognizable compared to LF3PFL. In contrast, the perturbed images from LF3PFL showed a high degree of distortion, indicating a robust resistance to DLG attacks and reflecting LF3PFL’s significant superiority in ensuring the privacy of training data.

These comparisons demonstrate that LF3PFL far exceeds other methods in terms of privacy protection, especially against privacy attack techniques such as DLG. The exceptional performance of LF3PFL, in terms of design and practical application, validates its effectiveness in safeguarding user privacy. Hence, LF3PFL’s performance in privacy-preserving federated learning algorithms can be considered the best among current methods, marking an important advancement for federated learning applications that handle sensitive data.

## 5. Conclusions

In conclusion, LF3PFL represents a significant advancement in the field of federated learning (FL), addressing the critical challenge of balancing data privacy with model utility. Our research highlights the innovative approach of LF3PFL in seamlessly integrating privacy-preserving mechanisms into existing FL frameworks without compromising the efficiency and accuracy of the learning process. The method’s robustness against sophisticated privacy attacks, such as deep leakage from gradients (DLG), underlines its potential to safeguard sensitive information in a variety of FL applications. Furthermore, the open-source availability of LF3PFL’s prototype implementation fosters a collaborative environment for further exploration and development within the research community. By offering this tool, we aim to catalyze advancements in both privacy protection techniques and federated learning methodologies.

Key insights from our research underscore LF3PFL’s adaptive excellence in diverse settings, proving its efficacy across different datasets and model architectures. This adaptability, coupled with the method’s seamless integration and enhancement of privacy without sacrificing performance, sets a new benchmark for future FL solutions. LF3PFL’s contribution to the field not only addresses current privacy and efficiency concerns but also opens new avenues for the practical application of FL in sensitive and critical domains, ultimately contributing to the broader goal of secure and efficient distributed machine learning.

## Figures and Tables

**Figure 1 entropy-26-00353-f001:**
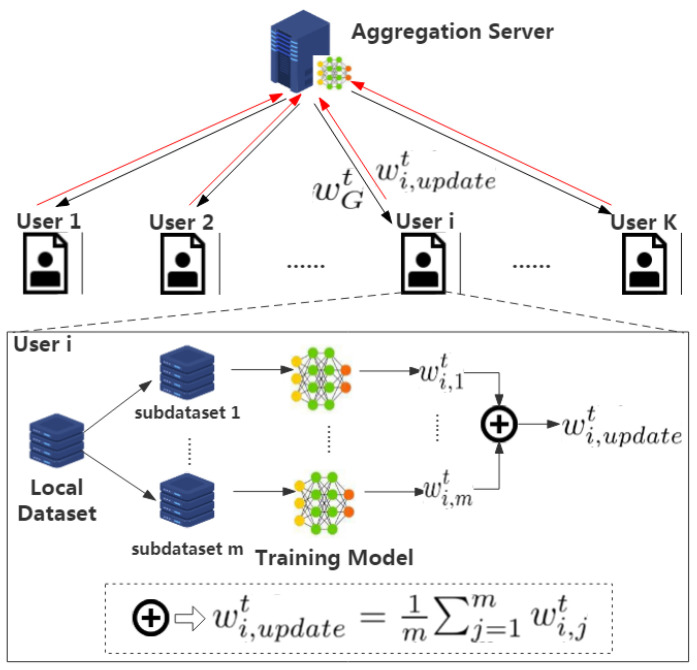
Architecture of the LF3PFL scheme.

**Figure 2 entropy-26-00353-f002:**
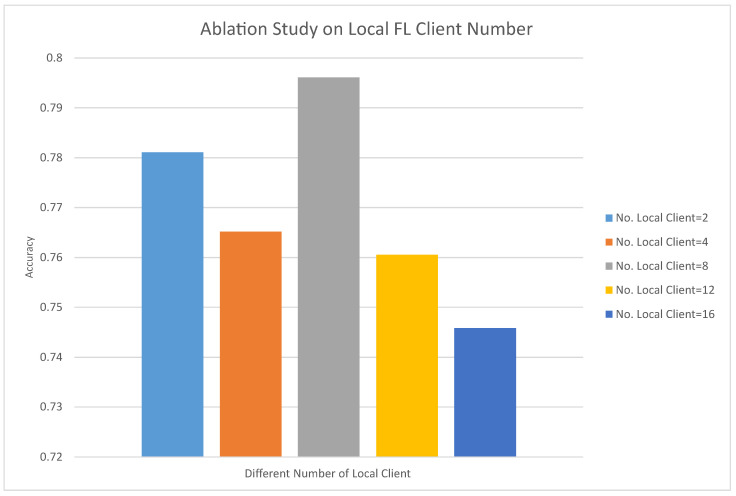
Accuracy of ablation study on different local FL client numbers. The local federated model used is ModerateCNN, and the test dataset is CIFAR-10.

**Figure 3 entropy-26-00353-f003:**
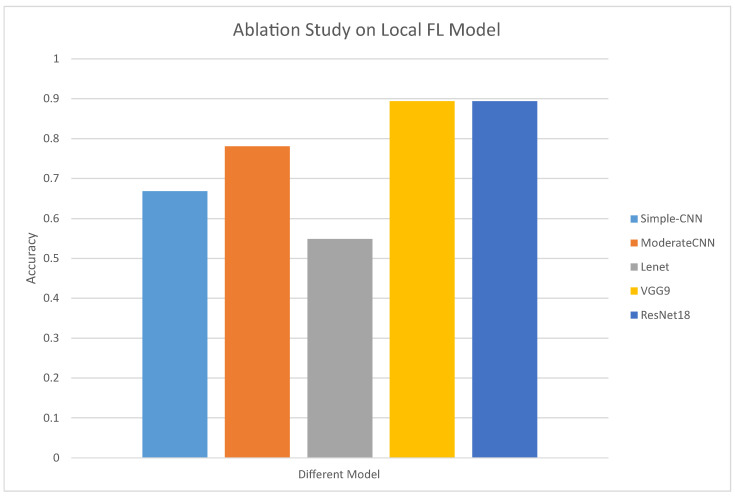
Accuracy of ablation study on different local FL models. The local FL client number is fixed to 2, and the test dataset is CIFAR-10.

**Figure 4 entropy-26-00353-f004:**
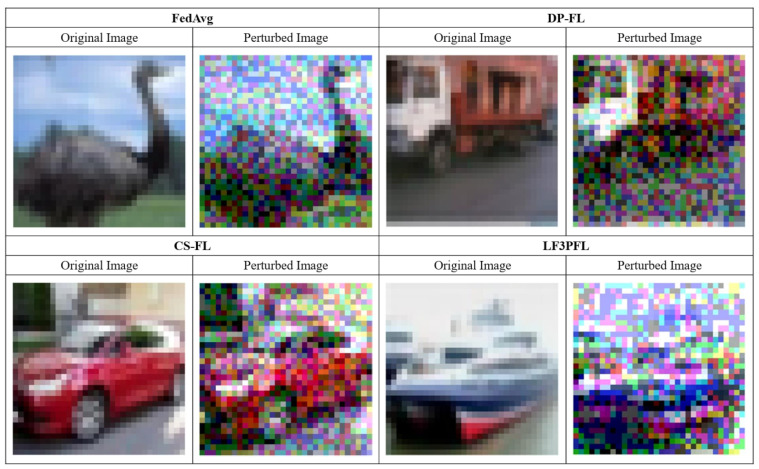
Privacy attack experiment result. In this comparative experiment, each row represents a different FL (federated learning) method, with the original image from the training data on the left and the image reconstructed using the DLG (deep leakage from gradients) attack on the right. The methods being compared include FedAvg, DP-FL (differential privacy federated learning), CS-FL (compressed sensing federated learning), and our proposed LF3PFL method.

**Table 1 entropy-26-00353-t001:** List of symbols and their meanings.

Symbol	Meaning
*K*	Number of clients in the federated learning system
*T*	Number of training rounds
*B*	Local minibatch size for training
*E*	Number of epochs for local training
η	Learning rate for the optimization algorithm
$w_{G}^{0}$	Initial global model parameters
$w_{G}^{t}$	Global model parameters in round *t*
$w_{k}^{t}$	Local model parameters of client *k* in round *t*
$w_{k,n}^{t}$	Local federalized model *n* of client *k* in round *t*
$D_{k}$	Local dataset of participant *k*
$D_{k,n}$	Subset *n* of the local dataset of participant *k*
*N*	Number of data subsets into which a local dataset is divided
*m*	Number of participants selected in each round for updating
ℓ(w;b)	Loss function evaluated on parameters *w* and batch *b*

**Table 2 entropy-26-00353-t002:** Comparative analysis of SOTA FL accuracies in various datasets.

Non-Privacy-Enhanced FL				Privacy-Preserving FL			
	**FedAvg**	**FedProx**	**FedMA**	**DP-FL**	**Chain-PPFL**	**CS-FL**	**LF3PFL**
Cifar-10	76.71	81.54	**82.12**	79.81	81.56	80.01	**81.63**
MNIST	89.46	84.69	**99.13**	81.88	85.65	82.92	**97.21**
Shakespeare	**56.24**	53.49	47.74	52.39	52.74	52.27	**55.86**

## Data Availability

Data are contained within the article.

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
