# Peer review of "LF3PFL: A Practical Privacy-Preserving Federated Learning Algorithm Based on Local Federalization Scheme"

_entropy, 2024, doi:10.3390/e26050353_

Round 1

Reviewer 1 Report

Comments and Suggestions for Authors

The article presents LP3PFL, a federated learning framework that addresses the challenges in federated learning to balance privacy protection and the model's efficacy. The authors compare their proposed method comprehensively with other state-of-the-art approaches and experiment on different datasets to demonstrate their model's generation ability to handle diverse datasets. However, the authors need to provide more details about the three stages, especially the data segmentation stage. It is imperative to understand how they determine the size of each subset and how they handle mapping the data points in the case of non-IID data. 

The paper also contains minor typos, such as "i.i.d" on page 7 and a text overwrite, "Chained-Communication mechanism" on page 10.

Author Response

Dear Reviewer,

Thank you for your constructive feedback and for pointing out the need for additional details regarding the three stages of our research, particularly the data segmentation stage. We appreciate the opportunity to clarify these aspects and enhance the completeness of our manuscript. Here, we provide detailed explanations of our methodologies used in determining the size of each subset and how we handle the mapping of data points for non-IID data scenarios (The corresponding revisions and detailed methodologies are documented in the "Data Segmentation" section on pages 7-8 of our manuscript. This section has been thoroughly updated to reflect these clarifications and ensure a comprehensive understanding of our approach.):

Data Segmentation for CIFAR and MNIST Datasets:

For the CIFAR-10 and MNIST datasets, we employed the Dominant Class Partition method to segment the data. In this method, the client data consists of a dominant class and other classes. We use a parameter Q to represent the proportion of the dominant class within the dataset. In our experiments, setting Q=0.5 means that 50% of the data belongs to one label, while the remaining 50% comprises random samples from other categories. This segmentation approach allows us to simulate a balanced scenario where the influence of any single class is moderated, providing a robust test environment for our algorithms under conditions where data is not identically distributed across classes.

Shakespeare Dataset:

For the Shakespeare dataset, in the non-i.i.d. sampling scenario, we maintained the distribution of data for each user consistent with the original dataset's composition. Given that we assume data distributions vary among users in the raw data, this type of sampling was categorized as non-i.i.d. This approach reflects the real-world scenario where data distribution often varies significantly from one user to another, which is crucial for testing the adaptability and efficiency of our proposed models in handling diverse data distribution patterns.

Corrections to Typos and Text Overwrites:

Regarding the minor typographical errors and text overwrites mentioned, such as "i.i.d" on page 7 and "Chained-Communication mechanism" on page 10, we have made the necessary corrections in the corresponding sections of the manuscript. These edits have been implemented to ensure the accuracy and clarity of the information presented, thus maintaining the high standards of our publication.

Reviewer 2 Report

Comments and Suggestions for Authors

In this paper the authors proposed a privacy-preserving federate learning algorithm, LF3PFL that enhances data confidentiality and model efficacy through localised federated updates. The paper follows good standard of research in this widely research field. The motivation and goals are clear, the algorithms are written formally and experiments are presented as the proof of concept. The result also validates the benefit of the algorithms. Some suggestions to improve the robustness and generalisability of the paper includes: (1) investigate the scalability of the proposal by using a larger number of local clients, (2) use varied datasets in the experiments.

Author Response

Dear Reviewer,

Thank you for your constructive comments on our manuscript, particularly your recommendations to (1) assess the scalability of our approach by employing a larger number of local clients and (2) incorporate a broader range of datasets in our experiments. We value your insights and recognize the importance of these aspects in enhancing the robustness and applicability of our research.

Regarding your first point, as detailed on page 9 of our paper, we set the number of local clients based on the number of baseline methods we planned to compare our approach against. This was designed to ensure a consistent experimental framework that allows for fair and direct comparisons with other existing algorithms. By using a fixed number of local clients, we were able to clearly demonstrate the specific impact and performance of our approach compared to established benchmarks. However, we acknowledge the critical importance of scalability and intend to expand our experiments to include a greater number of local clients in future studies. This will help us further validate and improve the scalability of our proposed method.

For your second point, our current study involved extensive testing using three widely recognized datasets, which enabled us to rigorously evaluate and showcase the efficacy of our algorithm under diverse conditions. These datasets were chosen for their relevance and frequent use in the field, ensuring a robust foundation for our initial investigations. Moving forward, we are committed to including a wider variety of datasets in our research. This expansion will not only aid in further refining our algorithm but will also enable us to examine its applicability and resilience across different data types and scenarios, thus significantly extending the reach and impact of our findings.